# Antithrombotic Therapy in Patients with Atrial Fibrillation and Acute Coronary Syndrome

**DOI:** 10.3390/jcm9072020

**Published:** 2020-06-27

**Authors:** Wilbert Bor, Diana A. Gorog

**Affiliations:** 1St. Antonius Hospital, 3435 CM Nieuwegein, The Netherlands; 2Department of Medicine, National Heart & Lung Institute, Imperial College, London SW3 6LY, UK; d.gorog@imperial.ac.uk; 3Postgraduate Medical School, University of Hertfordshire, Hatfield AL10 9AB, UK

**Keywords:** atrial fibrillation, acute coronary syndrome, antithrombotic therapy, percutaneous coronary intervention, oral anticoagulants, antiplatelet therapy, dual therapy, triple therapy, vitamin K antagonist, NOAC

## Abstract

Acute coronary syndrome and atrial fibrillation are both common and can occur in the same patient. Combination therapy with dual antiplatelet therapy and oral anticoagulation increases risk of bleeding. Where the two conditions coexist, careful consideration is needed to determine the optimal antithrombotic treatment to reduce the risks of future ischaemic events associated with both conditions. Choices can be made in intraprocedural anticoagulation, type and dosing of oral anticoagulant, duration of combination therapy, and selection of P2Y_12_ inhibitor including genetic testing. This review article provides an overview of the available evidence to support clinicians in finding the delicate balance between antithrombotic efficacy and bleeding risk in patients with acute coronary syndrome and atrial fibrillation.

## 1. Introduction

Acute coronary syndrome (ACS) and atrial fibrillation (AF) are both common and can occur in the same patient. Concomitant AF exists in 16% of patients presenting with myocardial infarction (MI), of which one third is newly diagnosed [1,2]. From the other perspective, up to 36% of patients with AF have concomitant coronary artery disease (CAD) amongst whom 45% have prior MI [3,4]. Treatment of ACS, with or without percutaneous coronary intervention (PCI), requires the administration of dual antiplatelet therapy (DAPT) to prevent recurrent MI and stent thrombosis (ST) [5,6]. In most patients with AF, oral anticoagulation (OAC) is indicated for the prevention of stroke and systemic embolism [7]. Thus, there is a difference in antithrombotic treatment requirements between patients with AF and those with ACS, and where the two conditions coexist, careful consideration is needed to determine the optimal antithrombotic treatment to reduce the risks of future ischaemic events associated with both conditions [8,9]. However, a combination of DAPT and OAC (also referred to as “triple therapy”), confers a significant risk for major bleeding, which is up to four times higher than that associated with OAC alone [10]. Since major bleeding after PCI has been shown to be associated with increased mortality [11], the bleeding risk should be kept to a minimum. Here, we aim to provide an overview of the available evidence and support clinicians in finding the delicate balance between antithrombotic efficacy and bleeding risk in patients with ACS and AF.

## 2. Comparison of Antithrombotic Therapy for ACS and AF

Aspirin has been the cornerstone of CAD treatment for decades, to which, after ACS, a P2Y_12_ inhibitor is added to further reduce major adverse cardiac events (MACE, namely cardiovascular death, MI, and stroke) [12]. In the setting of ACS, DAPT comprising of aspirin combined with the more potent P2Y_12_ inhibitors ticagrelor or prasugrel is preferred, as these further reduce MACE, compared to DAPT comprising of aspirin with clopidogrel [5]. However, this gain in efficacy with ticagrelor or prasugrel over clopidogrel is offset by a tradeoff in safety, with increased bleeding with these agents compared with clopidogrel [13,14]. The recent open label ISAR REACT 5 trial, in which patients with ACS were randomized to ticagrelor or prasugrel, showed that patients treated with prasugrel had fewer MACE than those on ticagrelor, yet with similar frequency of bleeding [15]. Furthermore, the randomized POPular AGE trial showed that in elderly (≥70 years) non-ST elevation ACS patients, without AF, clopidogrel reduced bleeding and was noninferior with respect to a net clinical benefit of death, MI, stroke, and bleeding, compared to ticagrelor [16].

For patients with AF with a CHA_2_DS_2_-VASc score of ≥2, treatment with OAC is indicated for the prevention of stroke and systemic embolism and is more effective than antiplatelet therapy [7,8]. The nonvitamin K antagonist oral anticoagulants (NOACs) apixaban, dabigatran, edoxaban, and rivaroxaban are preferable for eligible patients (i.e., in the absence of a mechanical heart valve, moderate-to-severe mitral stenosis, and severe renal insufficiency) over vitamin K antagonists (VKA), due to their superior safety and efficacy profile [7]. In particular, treatment with a NOAC is associated with a 50% lower risk of intracranial haemorrhage compared to VKA [17].

## 3. Trials of Dual or Triple Antithrombotic Therapy for PCI with or without ACS

To treat patients with both ACS and/or PCI and AF, studies have assessed the benefit of combination antithrombotic strategies comprising of both antiplatelet agents and OAC. The combination of OAC and DAPT (“triple therapy”) carries a significant increased risk of bleeding, compared to either DAPT or OAC alone, which is associated with excess mortality [10,11]. Therefore, several trials have evaluated the safety and efficacy of combination antithrombotic regimens dropping aspirin (“dual therapy”). An overview of the trials is given in Table 1. Of note, none of these trials had adequate power to detect differences in antithrombotic therapy efficacy for preventing ischaemic events. Godino et al. analyzed trials comparing double versus triple antithrombotic therapy in patients with AF undergoing PCI and showed that whilst the trials seem to indicate that double therapy significantly reduces the risk of bleeding, the trials were not designed or powered to address safety concerns with respect to the prevention of ischaemic events in patients with ACS, particularly those at high ischaemic risk [18]. The authors of that analysis recommend a personalized strategy, with a careful individual assessment of the patient ischemic and bleeding risk.

### 3.1. Aspirin versus No Aspirin

The first study to address this was the small open-label WOEST trial, which randomized 573 patients with various indications for OAC after PCI to VKA and clopidogrel with or without aspirin. The primary outcome of any bleeding complication at one year post-PCI was very significantly reduced in the dual compared to triple therapy group (hazard ratio [HR] 0.36, 95% CI 0.26–0.50, *p* < 0.0001), but importantly, the trial was insufficiently powered to assess safety in terms of ischaemic endpoints [19]. In the PIONEER AF-PCI trial, 2124 patients with nonvalvular AF undergoing PCI were randomized 1:1:1 to dual therapy comprising of reduced dose 15 mg rivaroxaban with a P2Y_12_ inhibitor for 12 months, very low dose rivaroxaban 2.5 mg b.i.d. plus DAPT for 1, 6, or 12 months or triple therapy with VKA [20]. The study showed that the use of either low-dose rivaroxaban plus a P2Y_12_ inhibitor or very-low-dose rivaroxaban plus DAPT was associated with fewer major bleeding events than VKA plus DAPT. In the RE-DUAL PCI trial, 2725 patients with AF who had undergone PCI were randomized to a dual therapy regimen of dabigatran 110 mg or 150 mg plus P2Y_12_ inhibitor or triple therapy with VKA plus DAPT [21]. Dual therapy with both dabigatran 110 mg and 150 mg significantly reduced the primary endpoint of major or clinically relevant nonmajor bleeding compared to the corresponding triple therapy group (110 mg dabigatran: HR 0.52; 95% CI 0.42–0.63; *p* < 0.001 for noninferiority; *p* < 0.001 for superiority; 150 mg dabigatran HR 0.72; 95% CI, 0.58–0.88; *p* < 0.001 for noninferiority). In the AUGUSTUS trial, 4614 patients with AF and either ACS or undergoing elective PCI were randomized in a 2 × 2 factorial design to aspirin or placebo and to apixaban or VKA, in addition to P2Y_12_ inhibitor [22]. Bleeding was lower with apixaban than with VKA (HR 0.69; 95% CI 0.58–0.81; *p* < 0.001 for both noninferiority and superiority) and triple therapy was associated with significantly more bleeding than dual therapy with OAC and a P2Y_12_ inhibitor (without aspirin) (HR 1.89; 95% CI 1.59–2.24; *p* < 0.001). More recently, the ENTRUST-AF PCI trial was reported, in which 1506 patients with AF and PCI for stable CAD or ACS were randomized to dual therapy with edoxaban plus P2Y_12_ inhibitor or triple therapy with VKA and DAPT [23]. Major or clinically relevant nonmajor bleeding was lower with dual compared with triple therapy (HR 0.83, 95% CI 0.65–1.05; *p* = 0.0010 for noninferiority, margin HR 1.20; *p* = 0.1154 for superiority).

A meta-analysis of the trials evaluating NOAC dual therapy versus VKA triple therapy confirmed the significant reduction of major or nonmajor clinically significant bleeding with dual therapy (risk ratio [RR] 0.64, 95% CI 0.52–0.80, *p* < 0.0001) [24]. Efficacy with regards to cardiovascular death and stroke was similar (RR 1.10 [0.86–1.41] and 1.00 [0.69–1.45], respectively) but offset by an increased risk of stent thrombosis (ST) (RR 1.59, 95% CI 1.01–2.50, *p* = 0.04). Specifically, none of the trials truly assessed patients at high ischaemic risk, namely those with ACS, prior stent thrombosis, long/complex stented lesions, bifurcation stents, stents within the left main stem, or last remaining conduit.

#### Aspirin versus No Aspirin in the ACS Subgroups

It is especially interesting to examine the results in the ACS subgroups of the trials, since these patients are at higher risk of thrombotic events compared to those undergoing elective PCI. Although bleeding and antithrombotic effects of dual versus triple therapy in the ACS subgroups were grossly similar to the total study populations (Table 1), it is essential to remember that these studies were not powered to assess safety in terms of ischaemic events, let alone ischaemic events in ACS patients. In fact, in the RE-DUAL PCI study, numerically more myocardial infarctions and thromboembolic events or deaths were seen with dual therapy containing dabigatran 110 mg b.i.d. than with triple therapy with warfarin, although this was not statistically significant and this signal for possible harm was not seen with dabigatran 150 mg b.i.d. [25]. Subgroup analysis of patients with ACS in the AUGUSTUS trial showed that compared with placebo, patients treated with aspirin had a higher rate of bleeding than placebo in patients with ACS treated medically (HR 1.49, 95% CI 0.98–2.26) or with PCI (HR 2.02, 95% CI 1.53–2.67) with no difference in outcomes for the composite of death or hospitalization and death and ischaemic events [26]. However, subgroup analysis by ACS-related PCI, of patients in the RE-DUAL PCI and AUGUSTUS trials revealed that dual therapy compared with triple therapy was associated with a significant reduction in ISTH major bleeding (OR 0.51, 95% CI 0.38–0.68, *p* < 0.001), an increase in myocardial infarction (OR 1.43, 95% CI 1.02–2.00, *p* = 0.04) with a trend towards a higher risk of stent thrombosis (OR 1.92, 95%CI 0.98–3.75, *p* = 0.06), and no difference in the incidence of all cause death or stroke [27].

To conclude, for most patients with AF with ACS undergoing PCI, dual therapy might be an effective treatment option to reduce bleeding events, but for patients at high ischaemic risk (such as those at high risk of recurrent MI or ST), triple therapy should be considered.

### 3.2. NOAC versus VKA

Only the AUGUSTUS trial was designed to evaluate whether NOAC or VKA is preferable as part of dual or triple therapy. Use of apixaban was associated with a significant absolute 4.2% risk reduction in bleeding compared to VKA during six months’ follow-up. This was consistent across dual or triple therapy groups and for the ACS subgroup [22,26]. Since the superior safety of each NOAC compared to VKA in the AF-PCI trials was similar, we consider this a NOAC class-effect. Therefore, the use of NOAC is preferable over VKA as part of combination therapy in patients with AF and ACS [28].

### 3.3. Dose of NOAC

According to the guidelines, for all NOACs the lowest tested dosage should be considered to prevent excess bleeding, with dose reduction as required by the licensing [5,28]. Whilst for apixaban and edoxaban, the dose remains the same in combination therapy as that used for AF alone, for rivaroxaban the dose in combination therapy is reduced to 15 mg o.d. rather than the conventional 20 mg o.d. used for AF, based on the results of the PIONEER-AF-PCI study [20], and the recognized dose-dependent increase in bleeding with rivaroxaban [29]. Whilst dabigatran is available at both 110 mg and 150 mg b.i.d. doses for stroke prevention and also shows dose-dependent bleeding when combined with antiplatelet therapy [21,30], the 110 mg dose used in dual therapy might not be as effective in the prevention of thromboembolic events, and therefore may not be the preferred choice in patients with high thrombotic risk [21,25]. However, inappropriate reduction in NOAC dose, such as that below the clinically indicated dose, should in general be avoided since it may increase thromboembolic risk [31].

### 3.4. Duration of Dual or Triple Therapy

No data specific data for ACS on the optimal duration of dual or triple therapy are available. Therefore, we must apply results for mixed elective and nonelective PCI to this subject. In the ISAR-TRIPLE trial, 614 patients receiving OAC (84% for AF) who underwent PCI (32% for ACS) were randomized to receive clopidogrel for either six weeks or six months in addition to OAC and aspirin [32]. During nine months of follow-up, there was no significant difference in the primary endpoint of the composite of cardiac death, MI, ST, ischemic stroke, and TIMI major bleeding (HR: 1.14; 95% CI 0.68–1.91; *p* = 0.63) and no difference in the secondary endpoints of combined ischemic events (cardiac death, MI, ST, ischemic stroke) or TIMI major bleeding. Landmark analysis after six weeks, however, showed more bleeding events in the group continuing clopidogrel for six months [32]. Therefore, based on this trial, a strategy of six weeks is preferable to six months of triple therapy.

Taking into account the promising results of dual therapy, but a signal towards higher risk of MI and ST in this group, it is crucial to remember that even in the dual therapy treatment arms an initial period of triple therapy was administered until randomization (median around two days, up to six days) [20,21,22,23]. Posthoc analysis of the AUGUSTUS data showed the antithrombotic benefit of triple therapy in the first 30 days after PCI or ACS but not after 30 days up to six months [33]. It showed an absolute risk reduction of 0.91% (95% CI 0.08–1.74%). Therefore, an initial period of 1–4 weeks of triple therapy may be warranted for patients at increased thrombotic risk and low bleeding risk [34].

## 4. Choice of Antiplatelet Agent

### 4.1. Choice of P2Y_12_ Inhibitor in Combination Antithrombotic Therapy

Most trials of combination antithrombotic therapy used clopidogrel, with only a small number of studies using ticagrelor or prasugrel (Table 1). There are no randomized trials comparing the different P2Y12 inhibitors head-to-head as part of combination antithrombotic therapy.

The only data specifically in ACS patients is derived from the TRANSLATE ACS study. It showed increased risk of any bleeding with prasugrel as compared to clopidogrel as part of both triple and dual therapy (39.0% vs. 24.4%, *p* = 0.003, and 26.7% vs. 19.7%, *p* not reported, respectively) [35]. No significant difference in MACE was observed (13.1% vs. 7.9%, *p* = 0.333, and 7.0% vs. 8.9%, *p* not reported, respectively).

Studies containing a mixed population of ACS and stable CAD patients showed similar results. Although not powered to assess this comparison, in the AUGUSTUS trial, patients receiving clopidogrel experienced less bleeding compared to those taking prasugrel or ticagrelor, with a similar rate of death or ischemic events (bleeding events 12.1% vs. 17.6% vs. 18.6%, and death or ischaemic events 6.9% vs. 5.9% vs. 6.1%, respectively) [22]. Verlinden et al. reported similar results when comparing clopidogrel to ticagrelor or prasugrel as part of triple therapy, where clopidogrel was associated with reduced bleeding (12.7% vs. 28.6%, *p* = 0.017) and similar major adverse cardiac and cerebrovascular events (MACCE) (18.3% vs. 19.0%, *p* = 0.91) [36]. In the RE-DUAL trial, patients receiving ticagrelor experienced more bleeding events compared to those taking clopidogrel (26.3% vs. 20.1%, HR 1.35, 95% CI 1.05–1.72), without reduction in death, thromboembolic events, or unplanned revascularization (18.7% vs. 12.9%, HR 1.34, 95% CI 1.00–1.92) [25]. These effects were consistent regardless of whether dabigatran 110 b.i.d. or 150 mg b.i.d. dose was used and whether as part of dual or triple therapy. The Japanese TWMU-AF PCI registry compared DAPT-approved low-dose prasugrel 3.75 mg/day, which is approved in Japan because of the increased risk of bleeding with antithrombotic agents in East Asians, with clopidogrel as part of triple therapy [37]. The registry data with one year follow-up showed no significant difference between prasugrel and clopidogrel with respect to the incidence of bleeding or MACCE.

To conclude, ticagrelor and prasugrel should be avoided as part of triple therapy, due to the excess bleeding risk compared to clopidogrel, without evidence for antithrombotic benefit. In patients at high risk for thrombosis, dual therapy may comprise more potent P2Y_12_ inhibition than clopidogrel; however, clear evidence is lacking, especially for the ACS population, with a significant increased risk for potential bleeding. If more potent P2Y_12_ inhibition is needed, slight preference for prasugrel might be considered for superior efficacy; however, this evidence is derived from patients not on OAC [15].

### 4.2. Genetic Testing

The POPular Genetics trial found that genotype-guided clopidogrel prescription after ST-elevation myocardial infarction (STEMI) without an indication for OAC showed equal MACE, but less bleeding, compared to standard treatment with ticagrelor or prasugrel [38]. A suggestion of benefit has also been described in patients with AF undergoing PCI [39,40]. Therefore, if available, before prescribing more potent P2Y_12_ inhibition, genetic testing might be reasonable to consider if equal antithrombotic efficacy can be expected with clopidogrel to reduce bleeding complications and in the absence of high ischaemic risk.

### 4.3. Dropping Aspirin or P2Y_12_ Inhibitor in Dual Therapy

No trials have evaluated the comparison of OAC in combination with aspirin versus OAC with a P2Y_12_ inhibitor. A 2016 expert consensus document recommended single antiplatelet therapy with P2Y_12_ inhibitor over aspirin because of the pivotal role of P2Y_12_-mediated signaling in thrombotic and inflammatory processes; the established clinical role of P2Y_12_ inhibitors in reducing ST; its favorable efficacy profile regarding prevention of stroke, MI, and mortality; and better gastro-intestinal tolerance. In poor metabolizers of clopidogrel, aspirin might be the more reasonable option, particularly in those at low risk of arterial thrombotic events [41].

## 5. Peri-Procedural Considerations for PCI

### 5.1. (Dis)Continuation and Bridging of OAC

The observational studies AFCAS and WOEST compared patients undergoing PCI with uninterrupted VKA or with bridging therapy over a 30-day follow-up [42,43]. The AFCAS registry found numerically less bleeding (12.1% vs. 18.6%, *p* = 0.07) and MACCE (3.8% vs. 6.2%, *p* = 0.25) with uninterrupted OAC. The WOEST study found similar bleeding events (19.1% vs. 17.4%, *p* = 0.51) and MACCE (1.7% vs. 3.4%, *p* = 0.48) with uninterrupted OAC. In the case of VKA as part of combination antithrombotic therapy, an international normalized ratio (INR) in the lower range is preferable to reduce bleeding risk, since higher INR values are shown to increase bleeding risk [44].

For NOACs, less data are available. Observational data suggest that both continuation and omittance one day preprocedure may be safe [45,46]. A study comparing patients with AF undergoing primary PCI for STEMI found no difference with regards to in-hospital major bleeding between patients admitted without OAC and those admitted on chronic VKA or NOAC treatment (13.2% vs. 13.0% vs. 11.6%, respectively, *p* = 0.57) [47]. Therefore, it seems safe to perform acute PCI if needed. If possible, withholding NOAC for 24 h might be considered to reduce bleeding risk; however, this is not supported by data for PCI. Importantly, since all OAC increases bleeding risk, use of the radial approach, over femoral access, is important in reducing bleeding complications.

### 5.2. Intraprocedural Anticoagulation

Intraprocedural parenteral antithrombotic agents were not tested specifically in patients with OAC. However, not all parenteral anticoagulants are created equal and we can extrapolate some findings about concomitant anticoagulation from ACS studies. The OASIS V trial compared fondaparinux and enoxaparin in patients with ACS, although the trial specifically excluded patients with a non-ACS indication for anticoagulation, such as AF. Whilst overall the trial showed that fondaparinux was similar to enoxaparin in reducing the risk of ischemic events at nine days, it substantially reduces major bleeding [48]. However, subgroup analysis of the 34% of patients who underwent PCI showed that catheter thrombus was more common with fondaparinux than enoxaparin (0.9% vs. 0.4%); this was largely prevented by using unfractionated heparin (UFH) at the time of PCI [49].

Current international guidelines recommend intraprocedural intravenous anticoagulation in all patients, with the exception of those on VKA, in whom no parenteral anticoagulation is needed if the INR is >2.5 [50,51]. This is based on one study showing similar MACE but more access site complications in patients with therapeutic range INR undergoing PCI with added UFH compared to no added UFH (3.2% vs. 4.1% and 11.0% vs. 5.1%, respectively) [52].

Little evidence is available showing that dabigatran only, without additional intraprocedural UFH, may not provide sufficient anticoagulation during elective PCI [53]. In a randomized study involving 50 patients undergoing PCI, postprocedural levels of prothrombin fragments 1 + 2 and thrombin-antithrombin complexes were elevated in patients receiving dabigatran, compared to standard intraprocedural UFH. Rivaroxaban, on the other hand, might provide sufficient anticoagulation during elective PCI [54]. In a study of 108 patients randomized to rivaroxaban or standard intraprocedural UFH, those receiving rivaroxaban showed adequate suppression of prothrombin fragment 1 + 2 and thrombin-antithrombin complexes compared to those receiving standard intraprocedural heparin. However, in the absence of trial data confirming that NOAC alone is safe, the ESC guidelines recommend additional parenteral anticoagulation during PCI to prevent catheter thrombosis [50,51]. In patients on NOACs, additional intraprocedural low-dose anticoagulation (e.g., intravenous enoxaparin 0.5 mg/kg or UFH 60 IU/kg) should be added irrespective of the time of the last administration of NOAC.

In patients taking uninterrupted OAC, use of glycoprotein IIb/IIIa inhibitors should be limited to bail-out cases to reduce bleeding risk [28,41,55].

### 5.3. Intraprocedural Antiplatelet Therapy

No data exist regarding patients on OAC receiving dual therapy with or without DAPT loading doses for PCI. In the AF-PCI trials, omittance of aspirin was after PCI and randomization (2–6 days after PCI) [20,22,23]. The 2018 expert consensus recommends a loading dose aspirin for all patients undergoing PCI [56]. Regardless of OAC status, antiplatelet therapy with aspirin and a P2Y_12_ inhibitor should be administered with loading as usual and a preference for clopidogrel as part of continued dual or triple antithrombotic therapy. Pretreatment with P2Y_12_ inhibitor should be considered in those undergoing invasive treatment as soon as the ACS diagnosis is made [5,28]; however, the specific benefits of pretreatment before angiography versus post-PCI treatment have not been explored in patients with AF on OAC. To reduce the risk of bleeding, one option is to postpone the administration of P2Y_12_ inhibitors to the time of PCI, when the anatomy is known [51].

### 5.4. Postprocedure Anticoagulation on the ICU

In patients with ACS who are in cardiogenic shock or needing ventilation on the intensive care unit, NOAC/VKA should not be restarted post-PCI but treatment continued with UFH [57]. This is because such patients frequently have multiorgan failure and absorption of oral agents may also be unreliable.

## 6. Balancing Risks

Compared to patients not taking OAC, all patients with OAC should be considered at increased risk of bleeding [10]. Thus, in every patient treated, bleeding risk should be minimized where possible. For all patients, radial access is preferred over femoral access. There should be a low threshold for routine use of prophylactic proton-pump inhibitors in patients taking dual or triple antithrombotic treatments [41].

Whilst the generalizations above are important universal guiders, bleeding and thrombotic risk should be assessed and balanced on a case by case basis. If bleeding risk prevails, antithrombotic therapy should be limited and shortened. If thrombotic risk prevails, antithrombotic therapy may be intensified (using a more potent P2Y_12_ inhibitor) or prolonged (triple therapy, prolonged dual therapy). Advanced age, prior bleeding, low body-weight, chronic kidney disease, and anaemia are factors increasing bleeding risk. Thrombotic risk is increased with advanced age, prior MIs/ST, ACS, extensive CAD, chronic kidney disease, diabetes, suboptimal stenting, greater stent length, small stent diameter, or bifurcation stenting [41]. Posthoc analysis of lesion characteristics in the PIONEER AF-PCI trial illustrated that these decisions should not be based on one single characteristic, since no major differences in efficacy could be found for high-risk coronary lesions [58]. Similar results were reported from the REDUAL PCI trial: no benefit from triple therapy was found for procedural of clinical complex cases [59]. No risk scores have yet been developed for the AF-PCI population, but scores for the general CAD or AF population like the DAPT, CHA_2_DS_2_-VASc, or HAS-BLED scores might be helpful for estimating bleeding or thrombotic risk [60,61,62].

### Elderly Patients

Patients with advanced age are at increased risk of both bleeding and thrombotic events, challenging the optimal balance [63]. Subgroup analysis of all AF-PCI trials found a consistent benefit of dual versus triple antithrombotic therapy and furthermore, a consistent benefit for NOAC versus VKA in patients older than 75 or 80 years of age, showing a significant reduction in bleeding, except for dabigatran 150 mg in patients >80 years (however, this is based on only eight patients), and edoxaban which also did not show significant bleeding reduction in the general population [19,20,21,22,23]. The antithrombotic benefit of NOAC over VKA was more pronounced in older patients [22]. A trend towards greater reduction in ischaemic events with triple versus dual antithrombotic therapy in older patients was reported in the PIONEER AF-PCI and AUGUSTUS trials, but this did not reach statistical significance [20,21,22]. A prespecified subgroup analysis of patients >75 years in the RE-DUAL PCI trial found that dual therapy which included dabigatran 110 mg was less effective in preventing death or thromboembolic events than VKA triple therapy, whilst dabigatran 150 mg was similar to VKA in efficacy in reducing ischaemic events but also in the incidence of bleeding [64].

In short, in elderly patients, dual antithrombotic therapy that includes a NOAC reduces bleeding compared to dual therapy including VKA. Superior efficacy compared to VKA has been shown for apixaban and rivaroxaban. Dual therapy is generally preferable to triple therapy, but the latter might be warranted for patients at high thrombotic risk.

## 7. Conclusions

Patients with concomitant AF and ACS are at higher risk of bleeding, due to the need for combined antithrombotic therapy. On presentation, DAPT loading should be considered in all patients. If patients with established AF have an indication for long term OAC, the P2Y_12_ inhibitor of choice is usually clopidogrel, with the exception of patients at high ischaemic risk where ticagrelor or prasugrel might be considered. Periprocedural DAPT and additional intraprocedural parenteral anticoagulation should be given to all patients, with the exception of those on VKA with INR > 2.5. For most patients, a regimen of dual antithrombotic therapy comprising of a P2Y_12_ inhibitor and NOAC is indicated and is associated with significantly lower bleeding risk than triple therapy. In the absence of specific trials in AF and ACS, and a signal for increase in MI and stent thrombosis with dual versus triple therapy, an initial period of 1–4 weeks of triple therapy should be considered and is recommended for patients at high thrombotic risk and those who are not at excessive bleeding risk. Potent P2Y_12_ inhibitors should in general be avoided due to excess bleeding risk, and genetic testing might be considered before escalation from clopidogrel to more potent agents. In every patient, careful consideration of thrombotic and bleeding risk is warranted, in order to reduce the risk of ischaemic and bleeding events.

## Figures and Tables

**Table 1 jcm-09-02020-t001:** Trials of dual and triple therapy.

Study (Year)	Design Follow-up	*N*	Type of Patients	Intervention	Comparison	Endpoints	Results				
						Primary Safety Endpoint	Total Population			ACS Subgroup	
						Secondary Efficacy Endpoint	DAT	TAT		DAT	TAT
WOEST (2013)	RCT Open-label 1 year	573	Various indications for OAC with PCI 69% AF 27% ACS	VKA + clopidogrel	VKA + clopidogrel + aspirin Aspirin 1 year	Any bleeding	19.4%	44.4%	*P* < 0.001	11.6%	26.7%
						Death, MI, stroke, TVR, ST	11.1%	17.6%	*P* = 0.025	8.7 %	9.3 %
PIONEER AF-PCI (2016)	RCT Open-label 1 year	2124	AF with PCI 52% ACS	Rivaroxaban 15 mg o.d. + P2Y_12_i 93% clopidogrel	VKA + P2Y_12_i + aspirin 96% clopidogrel Aspirin 1, 6 or 12 months	TIMI major + minor + CRNMB	16.8%	26.7%	*P* < 0.001	20.5%	27.1%
						CV death, MI, stroke	6.5 %	6.0 %	*P* = 0.75	6.8 %	8.3 %
RE-DUAL PCI (2017)	RCT Open-label 1 year	2725	AF with PCI 50% ACS	Dabigatran 110 mg b.i.d. + P2Y_12_i 86% clopidogrel	VKA + P2Y_12_i + aspirin 90% clopidogrel Aspirin 1 month with BMS, 3 months with DES	ISTH major + CRNMB	15.4%	26.9%	*p* < 0.001	14.7%	27.8%
						Death, MI, stroke, SE	15.2%	13.4%	*p* = 0.30	13.6%	9.5%
				Dabigatran 150 mg b.i.d. + P2Y_12_i 87% clopidogrel		ISTH major + CRNMB	20.2%	25.7%	*p* = 0.002	20.5%	27.1%
						Death, MI, stroke, SE	11.8%	12.8%	*p* = 0.89	7.2 %	8.7 %
AUGUSTUS (2019)	RCT Blinded6 months	4614	AF with PCI or ACS 37% ACS + PCI 24% medically managed ACS	Apixaban 5 mg b.i.d. or VKA + P2Y_12_i 93% clopidogrel	Apixaban 5 mg b.i.d. or VKA + P2Y_12_i + aspirin 92% clopidogrel Aspirin 6 months	ISTH major + CRNMB	9.0 %	16.1%	*p* < 0.001	8.2 %	14.1%
						Death, MI, stroke, ST, urgent revascularization	15.7%	13.9%	NR	8.0 %	7.4 %
ENTRUST (2019)	RCT Open-label 1 year	1506	AF with PCI 52% ACS	Edoxaban 60 mg o.d.+ P2Y_12_i 93% clopidogrel	VKA + P2Y_12_i + aspirin 92% clopidogrel Aspirin 1–12 months	ISTH major + CRNMB	17.0%	20.1%	*p* = 0.115	NR	NR
						CV death, MI, stroke, SE, ST	6.5 %	6.1 %	NR	NR	NR

Abbreviations: ACS: acute coronary syndrome; AF: atrial fibrillation; b.i.d.: twice daily; BMS: bare metal stent; CRNMB: clinically relevant nonmajor bleeding; CV: cardiovascular; DAT: dual antithrombotic therapy; DES: drug eluting stent; ISTH: international society on thrombosis and hemostasis bleeding criteria; MI: myocardial infarction; NR: not reported; OAC: oral anticoagulation; o.d.: once daily; P2Y_12_i: P2Y_12_ inhibitor; PCI: percutaneous coronary intervention; RCT: randomised controlled trial; SE: systemic embolism; ST: stent thrombosis; TAT: triple antithrombotic therapy; TIMI: thrombolysis in myocardial infarction bleeding criteria; TVR: target vessel revascularization; VKA: vitamin K antagonist.

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
