# Peer review of "Antithrombotic Therapy in Patients with Atrial Fibrillation and Acute Coronary Syndrome"

_jcm, 2020, doi:10.3390/jcm9072020_

Round 1
Reviewer 1 Report
To the authors
I enjoyed reading the manuscript, which is well written. Please consider following my comments.
General
-It is the review of AF and ACS, but while I was reading, some of the sentences is for AF and PCI. I know we need to show the data of AF and PCI because the data of AF and ACS is relatively scarce, but please clarify what authors are discussing, AF and ACS vs AF and PCI in each paragraph. Readers will be confused.
1 Introduction
-none
2 Comparison of antithrombotic therapy for ACS and AF
-Please cite recent Popular Age trial of clopidogrel vs more potent P2Y12 inhibitors (PMID: 32334703).
-ISAR React 5 trial is one of the trial of prasgurel vs ticagrelor. Please cite other RCTs, and their data are conflicting.
3.1 Trials of DAT or TAT
-Please add results such as HR in table 1 including subgroup analysis of ACS.
-Structure of paragraph is confusing. Please clarify what authors are discussing AF and PCI (including stable patients) vs AF and ACS (and mostly with PCI)
- Please discuss risk of stent thrombosis. Although a meta-analysis of patients with ACS and AF undergoing PCI also revealed a decreased risk of bleeding without an increase of ischemic events (PMID: 32310855), it did not analyze stent thrombosis and ACS/AF patients and the previous meta-analysis revealed DAT might increase the risk of stent thrombosis (Eur Heart J 2019, 40(46):3757-3767.)
3.2 For example, this review is for AF/ACS, but last sentence is for AF and ACS or elective PCI. It is very confusing for readers. Even thought the data is scarce, authors should expand discussion more of AF/ACS patients.
3.3 Same as above. This paragraph did not discuss AF and ACS.
3.4 Triple or dual should be decided based on ischemic and bleeding risk (Eur Heart J 2018, 39(3):213-260.) And ACS patients have high ischemic risk after PCI including ST. Authors should include more about AF/ACS patients rather than general AF/PCI patients.
4.1 I do not think ticagrelor and prasgrel should be avoided as part of dual therapy for ACS patients even though data is scarce. Again, it depends on ischemic/bleeding risk and I understood these meds should be avoided for triple therapy, but we cannot say ticagrelor and prasgrel should be avoided as part of dual therapy for ACS patients.
Please specific the target populcaiton “AF and ACS patients” or “AF and PCI patients including ACS and elective ”. Authors’ sentences are confusing.
5.1-5.4
As I read this paragraph, I did not understood what kind of procedures such as non-cardiac surgery. I think authors are talking about PCI, so it should be clarified “PCI peri-procedural managements”
5-2 I think ACT calculation is necessary especially for PCI even if INR is more than 2.5.
6 Is it still true to avoid omeprazole with clopidogrel?
Author Response
Response to Reviewer 1 Comments
General - The main comment of the reviewer is that sometimes mixed populations or even elective PCI is not enough separated from ACS or PCI in the setting of ACS.
Thank you very much for your time and thoughtfull comments. We indeed agree that sometimes we did generalise this too much, also since data is very scarce and the only data for ACS has to be deduced from a mixed elective and non-elective PCI population. We made some adjustments and added paragraph 3.1.1 to separately discuss the ACS subpopulations of the main RCTs in the field.
Paragraph 2 - The reviewer advises to cite the POPular AGE trial and more trials on prasugrel and ticagrelor.
Thank you for this advice. We cited the POPular AGE trial since it is indeed very relevant to this subject. To our knowledge the ISAR React 5, however, is the only randomised study comparing the potent P2Y12 inhibitors head-to-head, so we could not add more evidence on this subject.
Paragraph 3- The reviewer suggests adding Hazard ratios to results in table 1, and to distinguish more between elective PCI and ACS. Furthermore, the possible increase in stent thrombosis should be emphasized.
Thank you very much for this comment.
Indeed, the distinction between stable CAD and ACS was not always clear. We added a last sentence to the first part of paragraph 3 to elucidate the populations of the trials were mixed stable CAD and ACS, and separated the results for the ACS subgroups in a subheading 3.1.1. Also, in paragraph 3.2 the recommendation for elective PCI is removed. For the subject of paragraph 3.4 no specific data for ACS is available, we addressed this limitation now.
Since no hazard ratios are reported for the subpopulations of ACS, we cannot add these results to the table (as they will require Cox-PH analysis on patient level data).
The suggested meta-analysis by Gargiulo regarding the signal for increased stent thrombosis with dual therapy is indeed important, and mentioned at the end of paragraph 3.1 as “A meta-analysis of the trials evaluating NOAC dual therapy versus VKA triple therapy confirmed the significant reduction of major or non-major clinically significant bleeding with dual therapy (risk ratio [RR] 0.64, 95% CI 0.52-0.80, p<0.0001).[24] Efficacy with regards to cardiovascular death and stroke was similar (RR 1.10 [0.86-1.41] and 1.00 [0.69-1.45], respectively), but offset by an increased risk of stent thrombosis (ST) (RR 1.59, 95% CI 1.01-2.50, p=0.04).”.
Paragraph 4 - The reviewer points out that the evidence for P2Y12 selection in ACS patients on oral anticoagulants is scarce, and the recommendation should be clarified as it seems that we recommend to never use potent P2Y12 inhibition.
Thank you for this remark. Indeed little evidence exists for this subject on ACS patients. The paragraph 4.1 is reorganized and the recommendation has been made clearer. “very high’ in the conclusion is changed to “high” with regards to thrombotic risk patients that might be eligible for potent P2Y12 inhibition.
Paragraph 5 – The reviewer notes that this is only about periprocedural management in case of PCI, not with for example surgery.
Thank you for this advice, we changed the paragraph title to “Peri-procedural considerations for PCI”.
Paragraph 6 – The reviewer questions whether it is still true to avoid omeprazole with clopidogrel.
We agree with the reviewer and have removed the specific recommendations with regards to avoiding (eso)meprazole.
Reviewer 2 Report
I consider the present review well done and important for current clinical practice because many patients with ACS are in anticoagulants therapy for AF. Therefore this is a hotly debated topic.
I suggest just minor issues and to better highlight:
1) the concept that any of the 4 RCTs on DOACs and PCI (AUGUSTUS, the PIONEER-AF PCI, the RE-DUAL PCI, and the ENTRUST-AF PCI trial) was powered for ischemic events (mainly related to ST and myocardial infarction). As presented in a recent Table from the AJC: Antithrombotic Therapy After Percutaneous Coronary Intervention in Atrial Fibrillation, Cosmo Godino et al. May 17, 2020DOI:https://doi.org/10.1016/j.amjcard.2020.05.012, while the reported power for primary end point (major or clinically relevant non-major bleeding) was higher than 75% for all the studies, the post hoc power for ischemic events was always lower than 40%.
2) In addition, procedural characteristics as stent type, type and number of lesions treated (CTO, bifurcation, LMT) complete revascularization, and procedural success were not reported, even if they strong correlate with worse ischemic outcomes. Moreover, most of patients included in the 4 trial and treated by PCI were not affected by ACS which predisposes to higher thrombotic risk over time. Finally, since these trials were designed to test bleeding end points, they enrolled a population where the hemorrhagic risk was higher than the thrombotic one.
I suggest to add these concepts in the text.
Finally, in page 2, 3.1 Aspirin versus no aspirin,
I feel important to add in the text that the WORST trial is severely limited by the fact that less <500 patients were randomized. Therefore the conclusions must be rightly limited by this low number of patients treated without aspirin.
Author Response
Response to Reviewer 2 Comments
Thank you very much for your extensive reading and remarks.
The reviewer suggests to highlight the analysis of Godino et al regarding power of the AF-PCI trials.
Thank you for this suggestion. This is a very recent, interesting, and important analysis indeed. We referenced to it in paragraph 3.0 as follows:
“Of note, none of these trials had adequate power to detect differences in antithrombotic therapy efficacy for preventing ischaemic events. Godino et al. analysed trials comparing double versus triple antithrombotic therapy in patients with AF undergoing PCI and showed that whilst the trials seem to indicate that double therapy significantly reduces the risk of bleeding, the trials were not designed or powered to address safety concerns with respect to the prevention of ischaemic events in patients with ACS, particularly those at high ischaemic risk.[18] The authors of that analysis recommend a personalized strategy, with a careful individual assessment of the patient ischemic and bleeding risk.”
The reviewer addresses that it is not clear how many patients with high thrombotic risk such as chronic total occlusion, bifurcations stenting, left main disease etc were included in the AF-PCI trials., and that most of them were not ACS patients. Therefore, the bleeding risks were higher than thrombotic risks. This should be addressed in the text.
Thank you very much for this important remark. For the general ACS populations from the studies some data are available and represented in the manuscript. We agree that patients at high risk of recurrent MI or ST might be underrepresented in these studies, and therefore the higher thrombotic risk might require more antithrombotic treatment, in form of triple therapy or prolonged dual or triple therapy. We addressed this in section 3.1 as follows, as well as the last sentences of paragraph 3.1.1 and in the conclusion.
“Specifically, none of the trials truly assessed patients at high ischaemic risk, namely those with ACS, prior stent thrombosis, long/complex stented lesions, bifurcation stents, stents within the left main stem or last remaining conduit.”
The reviewer stresses that the WOEST trial was very small and that this should be addressed.
Thank you for this comment. We agree that this trial was small, but pivotal. We have amended our discussion of this trial in section 3.1 as follows:
“The first study to address this was the small open-label WOEST trial, which randomised 573 patients with various indications for OAC after PCI to VKA and clopidogrel with or without aspirin. The primary outcome of any bleeding complication at 1 year post-PCI was very significantly reduced in the dual compared to triple therapy group (hazard ratio [HR] 0.36, 95% CI 0.26-0.50, p<0.0001), but importantly, the trial was insufficiently powered to assess safety in terms of ischaemic endpoints.”